# Nanoparticles as Drug Delivery Systems: A Review of the Implication of Nanoparticles’ Physicochemical Properties on Responses in Biological Systems

**DOI:** 10.3390/polym15071596

**Published:** 2023-03-23

**Authors:** Azeez Yusuf, Awatif Rashed Z. Almotairy, Hanan Henidi, Ohoud Y. Alshehri, Mohammed S. Aldughaim

**Affiliations:** 1Irish Centre for Genetic Lung Disease, Department of Medicine, RCSI University of Medicine and Health Sciences, Beaumont Hospital, D02 YN77 Dublin, Ireland; 2Department of Chemistry, Faculty of Science, Taibah University, Yanbu 30799, Saudi Arabia; 3Research Department, Health Sciences Research Center, Princess Nourah bint Abdulrahman University, Riyadh 84428, Saudi Arabia; 4Department of Biochemistry, College of Medicine, Imam Mohammad Ibn Saud Islamic University (IMSIU), Riyadh 11564, Saudi Arabia; 5Research Center, King Fahad Medical City, Riyadh Second Health Cluster, Riyadh 11451, Saudi Arabia

**Keywords:** nanotechnology, nanoparticles, drug delivery systems, nanomedicine

## Abstract

In the last four decades, nanotechnology has gained momentum with no sign of slowing down. The application of inventions or products from nanotechnology has revolutionised all aspects of everyday life ranging from medical applications to its impact on the food industry. Nanoparticles have made it possible to significantly extend the shelf lives of food product, improve intracellular delivery of hydrophobic drugs and improve the efficacy of specific therapeutics such as anticancer agents. As a consequence, nanotechnology has not only impacted the global standard of living but has also impacted the global economy. In this review, the characteristics of nanoparticles that confers them with suitable and potentially toxic biological effects, as well as their applications in different biological fields and nanoparticle-based drugs and delivery systems in biomedicine including nano-based drugs currently approved by the U.S. Food and Drug Administration (FDA) are discussed. The possible consequence of continuous exposure to nanoparticles due to the increased use of nanotechnology and possible solution is also highlighted.

## 1. Nanotechnology: Current State of the Art

Nanotechnology is the intentional engineering and manipulation of particulate matter into a physical state of between 1 nm and 100 nm that can be rearranged or reassembled into nano-systems with improved function [1]. The emergence of nanotechnology and its application have put Ireland for instance, at the forefront of scientific research in the last decade [2]. Nanoparticles are the ultimate result of the technological modification of matter, and depending on their sizes, they are a few degrees larger than an atom consequence of the molecular processing of matter. As they possess enhanced characteristics such as auto-reactive stability and self-reassembly, they are easily adaptable and can be modified to achieve a specific characteristic or intended properties such as high surface area when compared to conventional substances [3,4].

Nanotechnology, as a relatively new branch of science, has gained attention in the last two decades and is rapidly expanding from the academic arena into the industry. Due to the possible advancements that can be achieved by nanotechnology, it has been estimated that nanotechnology will impact the global economy by about three trillion dollars by 2020 [5], making the field highly viable economically speaking. This could be attributed to the unique physicochemical properties of nanoparticles at the interface of chemistry, medicine, physics, and engineering.

The field of nanotechnology is one of the fastest-growing areas of scientific research and development, with significant advances being made in a range of applications. Currently, the state of the art in nanotechnology covers a wide range of areas, including electronics, energy, materials science, biomedicine, and more. In electronics, researchers are exploring the use of nanoscale transistors and other components to create smaller, faster, and more energy-efficient devices. In energy, nanotechnology is being used to develop new materials and devices for solar energy conversion, energy storage, and more. In biomedicine, nanotechnology is being used to develop new diagnostic tools, therapies, and tissue engineering strategies. Overall, the current state of the art in nanotechnology reflects a highly dynamic and rapidly evolving field, with many exciting new developments and applications yet to come.

Nanoparticles and nanomaterials are increasingly being explored for their potential applications in medicine. One of the most promising areas of application is drug delivery, where nanoparticles can be used as carriers to deliver drugs to specific cells or tissues in the body. Nanoparticles can be engineered to have specific surface properties that allow them to selectively target diseased cells while avoiding healthy ones, which can increase efficacy and reduce the side effects of drugs [6]. Additionally, nanoparticles can be designed to release their cargo in a controlled manner, allowing for sustained drug delivery over time [7]. Nanoparticles can also be used for diagnostic purposes, such as contrast agents in medical imaging, or the detection of specific biomolecules in biological samples [8]. In regenerative medicine, nanomaterials can be used as scaffolds for tissue engineering or as carriers for growth factors and other signaling molecules that promote tissue repair and regeneration. While the field of nanomedicine is still in its early stages, these and other potential applications hold great promise for improving the diagnosis and treatment of a wide range of medical conditions.

This review will briefly look at the contribution of nanotechnology in selected biological fields with a focus on medicine and the concept of nanoparticle-enabled drug delivery systems for disease treatment. While the popularity of nanoparticles in medicine will be explored, the imminent harmful effects due to the wide application of nanoparticles as well as the development of nanoparticle drug delivery systems (DSSs) in mitigating these effects will be explored.

## 2. Industrial Application of Nanotechnology

### 2.1. Food Industry

With the increasing awareness and demand for healthy food products, research has been devoted to devising tools for improving food shelf life and nutrient absorption. Nanotechnology as an enabling technology has been widely employed in achieving these fits in recent years for food preservation and delivery of nutraceuticals [6,7]. Nanoparticles are added to packaging materials to act as barrier molecules or as antibacterial agents and have displayed great promise [7]. One of the more widely utilised nanoparticle additives for this purpose is that of silver nanoparticle (AgNP) primarily due to silver’s innate antibacterial properties. AgNP can be added to food products in form of an edible biodegradable casing for food products, such as fruits, meat, and poultry, or included as an active ingredient in the polymeric matrix of the packaging material [8]. In fact, some studies have investigated the preservative effect of AgNP-containing packaging on asparagus [9], poultry meat [10], orange juice [11], and strawberries [12], all of which improved shelf life by inhibiting the activities of pathogens such as *E. coli*, *S. aureus*, moulds, and yeasts. In addition to AgNP, Zinc oxide (ZnO_2_) and titanium dioxide (TiO_2_) are effective against a wide variety of food pathogens such as *S. aureus*, *Salmonella typhi*, and *Klebsiella pneumoniae* [13]. Their used in the preservation of food items such as orange juice, strawberries, and liquid egg albumen, as documented [7]. In addition, TiO_2_ and ZnO_2_ have both been used as food additives for their whitening and UV-protective properties, respectively [13].

Nano-encapsulation is a well-established technique used in retaining and enhancing the release of functional nutrients and flavour in food items. Typically, these encapsulations are carbohydrate-based delivery systems made from starch, cellulose, chitosan, and dextrin that have been modified [14]. For example, phosphatidylcholine-based liposomes have been employed in the delivery of vitamin C, and this encapsulation is found to be more effective at maintaining the bioavailability of the nutrient likely through controlled release of the content when compared with free supplements administered orally [15]. Chitosan nanoparticles, in particular, have been shown to improve the stability and bioavailability of bioactive compounds in foods, such as curcumin and resveratrol [16]. Polymer-based nanoparticles, such as chitosan and poly-(lactic-co-glycolic acid) (PLGA), have been investigated for their ability to encapsulate and deliver bioactive compounds, such as antioxidants and vitamins, in food products [17].

Silica nanoparticles have been studied for their potential as food additives and carriers for bioactive compounds, due to their high surface area and low toxicity. For instance, silica nanoparticles have been used to deliver nutrients, such as iron, and improve the sensory properties of food products, such as beverages and sauces [18].

In general, nanotechnology in the food industry has made it possible to extend the shelf life of fresh food products in a cost-effective and pragmatic manner. While nanoparticles used for the improvement of nutraceutical delivery may be relatively non-toxic, nanoparticles used in packaging, such as AgNP, which may sometime leach into the main food product, with possible toxicity problems. While it has been estimated that up to 80 µg of AgNPs can be consumed daily [19], there are limited and conflicting reports on toxic effects of ingested AgNPs. There are some reports considering AgNPs to be safe for consumption without toxic effects, while others have reported significant toxicity upon AgNP ingestion (<125 mg/body weight) with an accumulation of the nanoparticle within organs such as the liver, kidneys, and small intestine [20,21,22]. While this may be a source of concern, it is worth considering if the level of AgNPs trialed in these studies is achievable through diet.

### 2.2. Cosmetic Industry

There is considerable usage of nanotechnology in the cosmetic industry with cosmetic manufacturers now including nanomaterials in their products for a variety of reasons. In the lucrative sunscreen industry, nanoparticles of zinc oxide and titanium dioxide are routinely added to sunscreen by virtue of their sizes, and they act as efficient filters of UV radiation without serious health hazards [23] or unsightly “white streaking” when the cream is applied due to the reduction in particle size. Liposomes prepared from varying lipid formulations of synthetic or natural lipids are also widely used in cosmetics such as ethosomes and transferosomes that are used to improve transdermal delivery of active cosmetic ingredients. The primary justification for the inclusion of liposomes in cosmetics is to enhance the transdermal delivery of cosmetic ingredients based on the ability of the liposomal lipid bilayer to fuse with cell membranes and alter the membrane fluidity for easy entry and delivery of liposomal content [24]. In addition, AgNPs are important ingredients in many cosmetic products as effective antibacterial agents such as in bathing products as active antibacterial ingredients, and because of AgNP activity against different yeast strains, they are also present in different dental products such as mouthwash and toothpaste [25,26]. A practical example is the already commercialised Silvosept mouth rinse (https://product.statnano.com/product/4502, accessed on 29 November 2019) and Royal Denta Silver toothpaste/toothbrushes (http://www.royaldenta.com/en/, accessed on 29 November 2019), which all contain AgNPs as a main active ingredient. Although nanotechnology is commonplace in the cosmetic industry, just like any other industry, the use of nanoparticles in the cosmetic industry is not without safety concerns due to the general repeated use of many cosmetics.

### 2.3. Nanomedicine

Nanotechnology was first conceptualised in medicine by Dr. Richard P. Feynman in the late 1950s, while describing the creation of molecular machines with atomic precision that can be used in engineering and medicine. He described the use of molecular mechanical machines that are capable of carrying out surgery or those that can permanently reside in the body for functional assistance of damaged organs [27]. Nanotechnology has strongly influenced the field of medicine, influencing how diseases are treated, particularly with the use of advanced drug delivery systems from both natural and synthetic compounds. For instance, researchers at the Wyss Institute of Harvard University developed a “nano-robot” that can specifically target cancer cells to deliver anticancer drugs [28]. Nano-robots that can treat cardiovascular diseases such as those that can engage in blood vessel repair by acting as artificial platelets [29], or those that can treat patients with coronary artery occlusion [30] are also in development. One of the most important applications of nanotechnology in medicine is in drug delivery systems. It is currently hypothesised that most conventional drugs have poor bioavailability and aqueous solubility limiting their absorption and retention within biological systems [31], as such significant efforts are being made to improve the efficiency of many traditional/conventional drugs.

Many nanoparticles are thought to have improved pharmacokinetic properties due to their physical nature and reduced size; they can target specific cells for selective action dependent on the particle type. These particles can easily penetrate target cells and accumulate into subcellular structures to modify cellular processes, which may be beneficial in the treatment of lifelong diseases such as diabetes, cancer, and kidney diseases [32]. As such, many of these nanoparticles have already been approved by the Federal Drug Administration in the United States for clinical use (Table 1). Nanoparticles that are popularly used in research for therapeutic purposes include encapsulated mRNA (siRNA) or DNA (in gene therapy), inorganic metal and metal complexes, or chemotherapeutic agents with pharmacologic abilities [33,34]. However, some of these nanoparticles do not easily traverse the cell membrane, requiring delivery systems to alleviate such difficulties. Thus, different nanoparticle delivery systems have been developed, some of which include liposomes, micelles, chitosan, and synthetic dendrimers [35,36,37,38]. The entrapment of both hydrophobic and hydrophilic drugs into liposomes is possible, and this helps to bypass the toxicity associated with anticancer drugs [39]. In particular, nanoparticles enabled delivery systems such as liposomes are well established for disease treatment such as in Doxil^TM^ (liposomal doxorubicin), which is approved by the FDA for the treatment of Kaposi sarcoma and ovarian cancer (Table 1). As such, liposomal encapsulation represents an effective route that enhances the drug therapeutic effect. In addition, modification of liposomes allows for passive or active tumour targeting (Figure 1). This effect facilitates an efficient drug payload into the malignant tumour cells, while the non-malignant cells become minimally impacted. Encapsulation of doxorubicin within the DPPC-based liposome enhances the cytotoxicity of the drug and at the same time suppresses the toxic side effects, thus improving the antitumoural therapeutic efficacy in comparison to conventional doxorubicin [40].

Nanoparticles have also found favour in the medical and diagnostic imaging of internal organs and tissues due to their interactions with mammalian cells owing to their modifiable physicochemical characteristics such as size, shape, optical, magnetic, and electronic properties [69].

Iron oxide and silica-based nanoparticles have been used to develop multifunctional imaging platforms such as MRI/optical dual-modal imaging, which possess several advantages over existing positron emission tomography (PET) and computed tomography (CT), both of which have radiation-related concerns [70,71,72]. Iron oxide is a magneto-responsive metal that is also biocompatible due to its degradable nature within biological systems. This in addition to its optical properties makes it a good imaging material for MRI. Iron oxide nanoparticles have been widely used as contrast agents for MRI. They are superparamagnetic in nature and can enhance the contrast in MRI images by altering the magnetic relaxation times of tissues [69]. This property has been used in clinical imaging for various applications, including cancer detection, inflammation imaging, and atherosclerosis imaging. Similarly, Silica-based nanoparticles have also been used as X-ray contrast agents for CT imaging [73]. The high X-ray attenuation of silica nanoparticles makes them useful for enhancing the contrast in CT images, which can be useful for detecting and monitoring a variety of diseases and conditions, such as cancer and inflammation. Kim et al. [71] described a silica-based nanoparticle with a paramagnetic shell containing a luminescent core. This unique nanoparticle offers the possibility for multimodal imaging by using the magnetic field of the MRI and the optical feature of the nanoparticle core. The paramagnetic shell can also be functionalized with peptides or moieties of interest. This offers the possibility of specifically targeting cancer cells. With the magnetic/optical properties of the nanoparticles and the magnetic field of the MRI, this technique offers a way to detect and monitor changes in living tissues for diagnostic purposes, without the need for using radioactive tracers that are typically used in a PET or CT scan. In addition to all these, nanoparticles, in particular, AgNPs, are used as coating materials in medical garments, wound bandages, medical implants, and devices as antibacterial [74]. Conventional disinfection only exerts a bactericidal effect that may not be effective after disinfection. On the contrary, AgNP coatings on medical devices and clothing materials remain effective against a wide range of bacterial strains a long as the nanoparticle is retained on the material surface. Nanotechnology seems to be playing a prominent role in medicine from its use as a therapeutic agent based on bactericidal properties to imaging and diagnostic purposes. These innovations are possible due to the properties that are peculiar to nanoparticles (see Section 3), and increased nanotechnology research in the medical field does not show any sign of slowing down. However, designing these nanoplatforms requires several key parameters that must be considered to ensure the platform is effective and safe for its intended use. Some of the major concerns include:Size and Shape: The size and shape of the iron/silica nanoparticles must be carefully controlled to optimize their performance and minimize any potential toxicity.Surface functionalization: The surface of the nanoparticles can be functionalized with various moieties, such as polymers, antibodies, or small molecules, to target specific cells or tissues. It is important to consider the stability, specificity, and efficiency of these functionalisation in the design of the nanoplatform.Core stability: Some of these nanoparticles such as iron oxide are known to be highly reactive, so it is important to ensure that the iron oxide core of the nanoparticles is stable and does not degrade or aggregate in biological systems.Biocompatibility: The nanoplatform must be biocompatible, meaning that it should not elicit an adverse reaction in biological systems, such as inflammation, toxicity, or immune response.Release kinetics: For therapeutic applications, it is important to consider the release kinetics of the payload from the nanoplatform. The release rate should be carefully controlled to ensure that the payload is delivered in a manner that is effective and safe.Targeting and accumulation: For therapeutic applications, it is also important to consider the targeting and accumulation of the nanoplatform in the desired tissue or organ. The nanoplatform must be able to selectively target and accumulate in the desired location in order to maximize its therapeutic efficacy.

## 3. Physiochemical Properties of Nanoparticles in Medicine

Nanoparticles have various properties that facilitate enhanced pharmacologic behaviour when compared with larger molecules. As such, significant efforts are being made in research modifying the nanoparticle size, shape, surface area, and surface chemistry to maximise their benefits for medical purposes.

Different nanoparticles such as gold nanoshells, liposomes, and micelles are synthesised in various ways, and the sizes and shapes of these nanoparticles can be controlled during the synthesis process based on the intended functionality. Nanoparticles can agglomerate into larger-sized particles during synthesis, which may enhance or indeed suppress the nanoparticle cytotoxicity depending on composition. The surface chemistry of nanoparticles can be modified by adding reactive groups or molecules such as antibodies to surfaces in targeted drug delivery systems (Figure 1 and Table 2).

**Figure 1 polymers-15-01596-f001:**
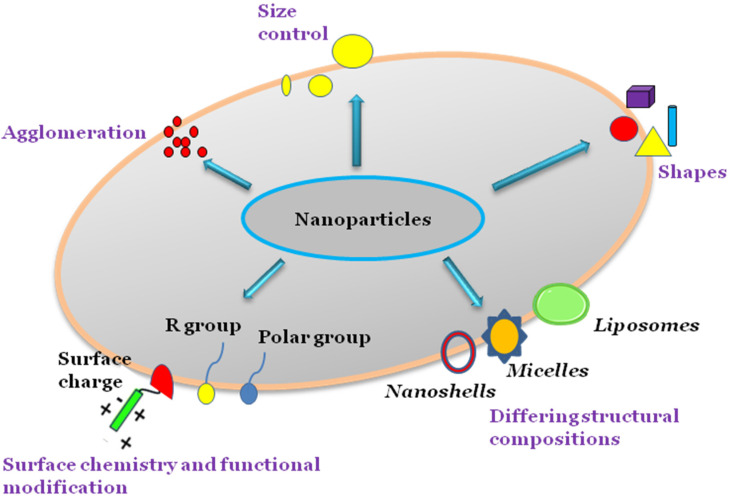
Physicochemical properties of nanoparticles.

Nanoparticles have different physicochemical properties including charged surfaces, the ability to agglomerate, the possibility of conjugating other groups to the surfaces, and controlled synthesis that facilitate specific shapes and sizes to be obtained. These properties allow nanoparticles to possess a more reactive nature in comparison to conventional particles within the biological environment.

### 3.1. Size and Surface Area

As stated, nanoparticles are small particles with sizes ranging between 1 nm and 100 nm, giving them a high surface area to volume ratio. By virtue of this property, nanoparticles have a high surface area of interaction per mass unit compared with more bulky particles, making some particles that are otherwise inert such as gold, to be reactive in the nanometer range [75]. A nanoparticle’s small size that is controllable also allows them to easily infiltrate body tissues and fluids, which are otherwise hindered when in the bulk form. In essence, the size and surface area of these particles contribute to the rate at which these nanoparticles are endocytosed, distributed, retained, and eliminated within biological systems [76]. As nanoparticles do not simply diffuse through the cell membrane, the extensive research into nanoparticles movements into normal and cancer cell lines has shown that they are internalised by endocytotic means in a size-dependent fashion [77,78]. Nanoparticles < 200 nm are known to be internalised by clathrin-coated vesicles, while larger nanoparticles, usually 500 nm, are known to be internalised by caveolae-mediated endocytosis [79]. In immune cells such as macrophages however, nanoparticles are prone to phagocytosis, and indeed research has shown that nanoparticles less than 500 nm in size enter immune cells through the phagocytotic pathway, while particles with larger particle sizes of between 2 and 3 µm, approximately around the size of bacteria cells, exhibit maximal phagocytotic uptake. Smaller nanoparticles such as liposomes can now be engineered for maximal uptake by mammalian cells based on their size [80]. For example, different lipid formulations can be used to prepare liposomes of specific sizes that can be more easily internalised by mammalian cells. Some studies that have used extrusion methods for instance, with a polycarbonate membrane of predetermined size to make liposomes of suitable sizes that can be easily internalised by mammalian cells. Such production methods of liposomes have been shown to improve the activity of chemotherapeutic drugs due to improved drug uptake by the cells [81].

In addition to uptake, intracellular localisation of nanoparticles has also been shown to be size-dependent. Oh et al. [82] showed that AuNP that were 2.4 nm in size were localised in the nucleus, while those larger than that up to 89 nm were localised in the cytoplasm after internalisation. Findings like these form the basis of how to modify nanoparticles size to evade or harness the immune system and how to localise nanoparticles in subcellular organelles of interest to maximise their effect. Furthermore, polymeric nanoparticles with a diameter ranging from 20 to 200 nm are effective for brain targeting and improving the penetration of drugs through the blood-brain barrier (BBB) [83]. In addition to size, the surface area of nanoparticles can also play a crucial role in their bioactivity. For example, PEGylated nanoparticles, which have a high surface area due to the presence of polyethylene glycol (PEG) on the surface, have been shown to improve the stability and circulation time of drugs in the bloodstream [84].

### 3.2. Surface Chemistry

The surface chemistry of nanoparticles such as charge or attached chemical groups is an important factor that determines their reactivity and ultimately can control their function. Many nanoparticles have been modified to change their surface chemistry to suit specific purposes. Rod-shaped gold nanoparticles (AuNPs) and DNA, because of their charge, cannot easily permeate or enter the cell. Both the AuNP and DNA have had their surfaces modified by coating them with lipid layers, while DNA has also been electrostatically conjugated to cationic liposomes to facilitate their transport into the cell, which resulted in improved uptake [80,85,86,87]. As liposomes and micelles have lipid layers that can interact and fuse with the cell membrane through hydrophobic interactions resulting in improved uptake, they can be used to deliver higher concentrations of nanoparticles intracellularly. Silicon nanoparticles (SiNPs) are important semiconductors that are used in optoelectronics, but their hydrophobicity hinders their application in biomedicine such as applications of internal imaging of tissues, since the biological system is aqueous and SiNPs are not stable in aqueous environments. Pan et al. [88] described a method of modifying the SiNP surface by coating it with silicon dioxide (SiO_2_) to make it more hydrophilic allowing for more biocompatible applications. ZnO_2_ nanoparticles are widely used in suscreen, due to their UV protection properties, but some studies have indicated the potential cytotoxicity of the nanoparticles making their application in cosmetic products worrisome. To negate this, some researchers have altered the surface properties, and indeed, a study has shown that by surface coating ZnO_2_ with poly methyl acrylic acid (PMAA), the cytotoxicity was reduced the nanoparticles retained their UV protection characteristics [89].

Liposomes are made up of phospholipids that mimic the lipid bilayer of the plasma membrane. The phospholipids component of the liposomes is amphiphilic with a polar head and a hydrophobic tail (Figure 2). The polar head is comprised of a phosphate group and glycerol both containing oxygen that can form hydrogen bonds in an aqueous environment. The hydrophobic tail on the other hand is made up of long-chain fatty acid, which aligns with the hydrophobic tail of another adjacent phospholipid, creating a hydrophobic core that can hold non-polar hydrophobic drugs in the bilayer so formed. The compatibility of liposomal surface chemistry with that of plasma membrane allows the adsorption of the liposome to the cell membrane where the liposome is internalised via receptor-mediated endocytosis or through fusion with the plasma membrane inducing membrane invagination and internalisation [90].

The pH of the environment where the particles are delivered can also affect the function of the nanoparticle-based on its surface chemistry, and this phenomenon has been utilised to trigger drug release in the tumour microenvironment that is characterised by acidic pH. For example, carrageenan oligosaccharide-capped AuNP have been recently shown to significantly release epirubicin in an acidic pH inducing cell death in HCT-116 colorectal cancer cells [91]. The surface of nanoparticles can alter their movement within aqueous biological systems and subsequently affect their reactivity or delivery. Such surface properties facilitate their use in a variety of ways such as in biomedical sensors, coatings of medical implants, and drug delivery systems. For example, an AgNP functionalized titanium implant surface was developed to prevent postoperative infection due to resistant strains of *Staphylococcus epidermidis* and *Staphylococcus aureus* due to the antimicrobial properties of AgNPs [92].

Liposomes are made up of a lipid bilayer with an outside aqueous core, both of which can be used for drug transport in disease treatment. Cholesterol is often added to the recipe for preparing the liposomes to restrict the fluidity of the phospholipid as in the plasma membrane.

As stated earlier, nanoparticles such as AuNPs and even conventional drugs are often coated with lipid layers as in liposomes to enable compatibility with the mammalian cell membrane, improving intracellular delivery. The liposome offers other functional benefits due to the active phospholipid heads that can be conjugated to a variety of compounds for targeted delivery. For instance, the conjugation of PEG to the surface of liposomes has been used to improve their bioavailability by making them undetectable to phagocytes, which eliminates them from the system [93]. PEGylation or the use of other linkers also facilitates the addition of active groups such as folate and monoclonal antibodies to the liposomal surface for selectively targeting specific cells (Figure 2). Folate is often used because of the high expression of folate receptors on cancer cell surfaces, which are used by the cancer cells to bind folate within the body for their uncontrolled proliferation [94,95]. On the contrary, monoclonal antibodies represent a more flexible approach for targeted delivery due to the countless number of unique receptors or surface antigens against which antibodies can be developed. Conjugation of these active surface agents to nanoparticle surfaces facilitates the delivery of the nanoparticles or drugs to the tumour cells for selective cancer cell eradication [96].

### 3.3. Shape

As stated previously, nanomaterials have tunable sizes, but their shape is also controllable during their synthesis. The shapes of nanoparticles can be altered during the last synthesis stage and typically involves nucleation of the nanoparticles from seed. The nucleation process involves the fusion of nanoparticle nuclei known as the seeds forming a template on which the nanoparticle crystals grow. Just like the size, the shape of a nanoparticle is paramount to its biological function and reactivity. Generally, nanoparticles that are round or spherical in shape are easily endocytosed in comparison to rod or tube-shaped nanoparticles [97]. This is because the shape affects endocytosis, which interferes with the way the membrane wraps over the nano-construct during contact. As such, the reduced endocytosis of nano-rods or other shapes is most likely due to the inability of the cell to initiate the necessary actin-dependent membrane kinetics required for endocytosis. This reason may explain why most nanoparticles with pharmacologic properties are spherical in nature. On the contrary, there are reports from new studies on nanoparticles of different shapes with potential applications in drug delivery. Zhao et al, [98] reported that in addition to their ability to encapsulate more particles, long-rod nanoparticles have prolonged bioavailability when compared to both spherical and short-rod nanoparticles. Other shapes such as nanoflowers and nanoprisms do exist, but these structures may not be as active as nanorods and nanospheres primarily due to their unique shapes [99].

The shape of a nanoparticle can play a significant role in determining its endocytosis. Endocytosis can occur through either clathrin-mediated endocytosis or clathrin-independent pathways (Figure 3). The shape of the nanoparticle can affect which endocytic pathway it takes, and thus, how it is internalized by the cell. Spherical or spherical-like nanoparticles tend to be taken up by clathrin-mediated endocytosis, which is a highly selective process [100]. The nanoparticle must fit specific size and shape criteria for clathrin to wrap around the particle and initiate internalization. However, nanoparticles with irregular or complex shapes are often internalized through clathrin-independent pathways [101]. This process is less selective and can be influenced by various factors such as particle size and charge. Additionally, certain shapes, such as rod-like or bristle-like nanoparticles, can become entrapped in the cell membrane and not be internalized by either clathrin-mediated or clathrin-independent pathways.

## 4. Nanoparticle Cytotoxicity

With the advent of nanotechnology and its growing application in almost all facets of everyday living, comes the concern on possible hazards resulting from increased human exposure. Significant research into the toxic effect or toxicity of nanoparticle exposure gave rise to the field of nanotoxicology. In recent years, this field has identified that the properties of nanoparticles that confer them with suitable pharmacologic behaviour are also responsible for their toxicity [102].

Several studies have investigated the toxicity of different nanoparticles using different cell lines and experimental conditions. For instance, toxicity of carbon nanotubes has been shown to affect the diversity of soil bacteria, [103], inhibit the growth of *Daphnia magna*, *Chlorella vulgaris*, and *Oryzias latipes* [104], and result in oxidative stress, membrane damage and inflammation in human A549 lung carcinoma cell line [105]. Different findings have shown that the mechanism of nanoparticle size-dependent cytotoxicity is due to their ability to infiltrate body tissues and subsequently enter cells to modify crucial cellular functions, one of which is to rupture the membrane of subcellular structures and induce the overproduction of reactive oxygen species (ROS) [106]. The presence of elevated levels of ROS induces oxidative stress that affects the normal physiological processes of the cell subsequently resulting in DNA damage, dysregulation of cell signaling, and ultimately cell death.

Nanoparticles are often surface-modified to enhance their functions. This may inadvertently result in increased cytotoxicity of the nanoparticle due to the influence of nanoparticle surface chemistry on its inherent toxicity. Based on localisation within the biological system, nanoparticles with reactive surface moieties can react with different intracellular or extracellular biomolecules; therefore, disturbing the normal processes needed to maintain tissue or cellular homeostasis. For instance, charged AuNPs have been shown to be more cytotoxic compared to neutral AuNPs as they induce higher levels of oxidative stress resulting in reduced mitochondrial function and increased expression of DNA damage-related genes [107]. Anionic cyanoacrylic nanoparticles are known to be more cytotoxic to macrophages compared with the cationic forms [108]. These differences may be a result of the macrophage’s phagocytotic affinity towards the bacterial cell membrane, which demonstrates an overall negative charge due to the Lipid A molecule of the LPS component of the bacterial cell membrane. Contrary to this, aminated iron oxide nanoparticles with an overall positive charge have been shown in a Chinese Hamster Ovary (CHO-K1) cell line to be efficiently internalised and hence induce higher cytotoxicity when compared to a PEGylated form [109]. Often, nanoparticles are PEGylated to increase their bioavailability and to reduce immunogenicity for a prolonged effect in vivo [84]. The retention of PEGylated particles in addition to the stability it confers them with may be a result of the slower uptake by cells resulting in a substantial reduction in their cytotoxicity.

Coupled with the size, the shape and aspect ratio of nanoparticles play a crucial role in their cytotoxic effects in vivo. It is believed that the higher aspect ratio of nanoparticles correlates with increased cytotoxicity due to reduced clearance and increased bioavailability of nanoparticles [110]. Higher aspect ratio nanoparticles often have cytotoxicity profiles similar to that of asbestos. Such particles can induce macrophage cell death during phagocytosis and, as was the case with asbestos fibers, have the potential to promote cancer development [111]. In support of this, Wozniak et al. [99] showed that gold nano-spheres and nano-rods, both of which were under 50 nm, were more cytotoxic on HeLa and HEK293T cell lines than their nano-star, nano-flower, and nano-prism counterparts that were all above 200 nm in size. It was postulated that this was because of a more efficient internalisation of these nanoparticles by the cells coupled with their optimum surface area for interaction with intracellular molecules.

## 5. Nanoparticle Drug Delivery Systems (DSSs) in Disease Treatment

Nanoparticles used in drug delivery range from 10 to 1000 nm in size with at least one dimension being below 100 nm in size. The small sizes of nanoparticles as well as their surface chemistry are known to offer pharmaceutically beneficial attributes but may also contribute to their toxic effects as discussed earlier. Smaller nanoparticles enter cells more effectively when compared with larger molecules, but the administration of nanoparticles with a reduced clearance may result in some of the particles being retained within the body. In the case of a more active or cytotoxic nanoparticle being retained rather than a bulk of the drug being eliminated during the first pass effect, this may result in harmful effects on the targeted site due to unwanted retention. Systemic administration of cytotoxic drugs may cause the drugs to exert their cytotoxicity on tissues during the first pass before they reach the intended tissues. Overall, 70% of globally synthesised drugs have poor aqueous solubility and hence poor pharmacokinetic properties in vivo [31]. As a solution to this, nanoparticle drug delivery systems (DSSs) have been developed to achieve targeted and more efficient delivery of the therapeutic substance, which would prevent damage to surrounding organs from the effect of administered drugs that will otherwise arise if the drugs were in the free form. Over the past few decades, research efforts into DSSs have advanced significantly with various DSSs already being investigated and developed for the treatment of diseases, such as cancer and neurodegenerative diseases [112,113]. Some of the typical nanoparticle DSSs currently under study are discussed below.

### 5.1. Lipid-Based DSSs

DSSs made from lipids vary in formulation and size and mainly consist of two types: namely micelles and liposomes. Micelles are formed through the self-assembly of a monolayer of lipid molecules in an aqueous environment into a nano-vesicle of between 5 and 50 nm [114]. They are used to successfully transport hydrophobic molecules, trapped in the hydrophobic core, at concentrations above their inherent water solubility. This is possible because the hydrophilic phospholipids are exposed to the aqueous environment while the hydrophobic tails form the core that can interact with the drug.

Unlike micelles, liposomes are bilayer nano-vesicles similar to the cell membrane with sizes ranging from 10 nm to several microns. The hydrophilic phospholipids of the outer layer are exposed to the aqueous environment, while that of the inner layer encloses the aqueous core (Section 3.2). Consequently, the hydrophobic tails of the bilayer lie above each other and are often used to trap hydrophobic drugs while the aqueous core is used to entrap hydrophilic drugs [90]. Liposomes have been one of the most useful tools in drug delivery in cancer treatment due to their ability to transport both water-soluble and insoluble drugs [115,116,117]. Conventional drugs, which are often small molecular drugs that have poor selectivity for tumour cells, are not retained within the tumour microenvironment as they diffuse back into the circulation system, causing cytotoxic side effects to normal cells. Liposomes, however, can improve the delivery of such drugs to the tumour microenvironment, which have tight junctions with gaps between 100 nm and 800 nm unlike normal epithelial junctions, which are 5 nm to 10 nm, via an enhanced permeability and retention (EPR) effect. Using the EPR effect, liposomes accumulate at the tight junctions of tumour cells and extravasate the blood vessels to the tumour microenvironment for delivery of the encapsulated drugs [118].

Liposomes generally have short half-life, but advancements in drug delivery research such as PEGylation of liposomes has allowed the development of liposomes with increased half-life [93]. In forming DSPE-PEG for instance, PEG is conjugated to phosphatidyl ethanolamine of DSPE via covalent linking of the amide group of DSPE to the carboxyl end of PEG [119]. Through the PEG linker on the liposome surface, several other moieties can be conjugated to the liposome as in targeted drug delivery. For example, click chemistry can be used to couple an azide-functionalized antibody to a Dibenzocyclooctyne-amine (DBCO)-PEG functionalized liposome in an azide–alkyne cycloaddition reaction [120]. Thus, liposomes can now improve both the pharmacodynamic and pharmacokinetic properties of water-soluble and insoluble drugs by increasing their delivery to target cells with potentially controlled release, thus lowering their cytotoxic effect on surrounding cells. For instance, liposomal doxorubicin coated with PEG (Doxil^®^) has been used to reduce doxorubicin cytotoxicity, improve its bioavailability and to enhance the delivery of the anticancer drug in different cancer types [121,122]. While PEGylation of liposome result in prolonged bioavailability by increasing the liposome hydrophilicity and reducing glomerular filtration/excretion, it often results in reduced uptake of the liposome by cells and degradation for drug release by the endo-lysosomal pathway [123]. Liposomes are widely used in targeted delivery due to the flexibility of their surface chemistry, which allows conjugation of targeting biomolecules such as peptides and antibodies that bond with specific cell surface receptors. This feature makes it possible to specifically target cells, such as cancer/tumour cells that express or overexpress specific receptors that recognises such molecules on the liposomes for targeted drug delivery (Figure 2 and Figure 3) [124].

Liposomes can be used to transport hydrophobic drugs in the lipid bilayer via hydrophobic interactions with the fatty acid tail of the phospholipids, while hydrophilic molecules, such as DNA or crystalline drugs, can be encapsulated within the aqueous core. Surface modifications are now possible on the surface of the liposomes allowing enhanced bioavailability, as occurred with PEG. Surface coating of drugs via electrostatic or ionic interactions or conjugation of antibodies, chemotherapeutic agents, peptides, and other proteins can prove useful for targeted delivery are routinely done with the aid of different linkers such as avidin-biotin complexes, PEG, or peptide linkers that are chemically conjugated to the phospholipid head and to the drug or protein of choice (Figure 2 and Figure 3).

**Figure 3 polymers-15-01596-f003:**
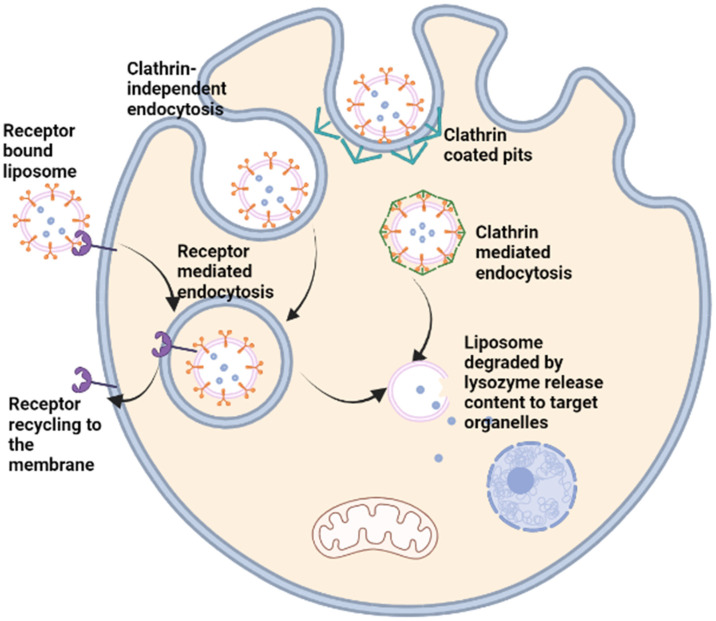
Targeted delivery and metabolism of liposome encapsulated drug.

In targeted drug delivery, liposomes are surface-modified by conjugating them with different groups using linkers such as PEG or chemical reactions such as the “Click Chemistry”(see Section 5.1). Such groups include drugs, peptides, or antibodies specific for cell surface receptors that are overexpressed by cancer cells for example, which facilitate the binding of such groups to the receptors and prevent access to normal cells. The liposome can be internalised through clathrin mediated endocytosis, through invagination of the cell membrane as endosomes in a clathrin-independent pathway, or through receptor-medicated endocytosis in which the receptor is recycled and returned to the cell membrane. Drugs encapsulated in the liposome are released to subcellular structures after degradation of the endosome and the liposome by the lysozymes in the endo-lysosomal pathway.

Lipid nanoparticles (LNPs) have been shown to effectively deliver nucleic acid-based therapeutics to target cells. LNPs can be engineered to include a variety of lipids, including cationic lipids, neutral lipids, and cholesterol, to optimize their stability, encapsulation efficiency, and cellular uptake. Additionally, LNPs can be functionalized with targeting ligands to improve specificity for certain cell types or tissues. Several recent studies have investigated the use of LNPs for delivering gene therapy to treat various diseases. For example, Nance and Hakim [103,104] reported the use of LNPs to deliver CRISPR-Cas9 for the treatment of muscular dystrophy has been demonstrated for the treatment of Duchene’s muscular dystrophy [103]. Engineered LNPs with a cationic lipid and PEGylated lipid are used to optimize their stability and cellular uptake. These LNPs are loaded with dystrophin mRNA as well as the CRISPR-Cas9 machinery and injected intravenously into mice with muscular dystrophy to effectively deliver the gene therapy to muscle cells, resulting in improved muscle function and reduced symptoms of the disease.

In addition to gene therapy, LNPs have also been investigated for delivering gene editing tools, such as CRISPR-Cas9 and base editors. LNPs have been trialed for delivering base editors or a CRISPR-Cas9 system that can delete the BCL11A gene to treat β thalassemia and sickle cell disease in human subjects [105]. The result showed a long-term deletion of the deleterious alleles in hematopoietic stem cells can an increase in the expression of foetal haemoglobin can reduce vaso-occlusive events. The use of LNPs for delivering gene therapy and editing has shown promising results in preclinical studies, with potential applications in treating various diseases. Further research is still needed to optimize LNP-mediated delivery for clinical translation.

### 5.2. Polymeric DSSs

Polymer-based nanoparticle DSSs are made up of a repeating unit of specific polymers and have been widely investigated for medical purposes in recent years [125,126]. Some of the known polymeric DSSs are PEG, chitosan, poly-(lactic-co-glycolic acid) (PLGA), and polylactic acid (PLA), but PEG, PLGA and PLA are the more widely studied, while chitosan research is beginning to gain more attention due to its biocompatibility, low immunogenicity and low toxicity [127]. Several PEGylated drugs have been approved by FDA for clinical use, making it the most commercialised polymeric DSSs (Table 2). PLGA and PLA are, however, known to be characterised by an initial burst release of the encapsulated drug (within 24 h) irrespective of the drug localization, and this may result in high delivery of drugs at unwanted sites, reducing drug benefits [128]. This has led to the development of polymeric DSSs with different triggers for the release of entrapped drugs. For example, some polymer-based DSSs are modified to respond to a subtle change in pH or ROS generation within the biological system. The pH or ROS level within a tissue may signal its physiological condition since ischemic or tumour tissue sites have higher pH or ROS levels when compared to normal tissue. Self-assembling and pH-sensitive polyamines have been demonstrated to possess flexible delivery capabilities. Doxorubicin-entrapped polyamine coated with folate and ligands for HIV transcriptional transactivator (TAT) were shown to be successful in the treatment of multidrug-resistant cancer cell lines [129,130]. These polyamines are pH-sensitive and release the entrapped doxorubicin in an acidic pH environment. Since the tumour microenvironment is mostly acidic and without the need for specific targeting molecules, this system can effectively target the tumour microenvironment. Dextran nanoparticles containing linked arylboronic esters that are degradable by ROS have also been used to deliver ovalbumin to murine dendritic cells to enhance their antigen-presenting abilities [131]. Because of the varying conditions that ensue in the tumour microenvironment, polymer-based DSSs seem to be flexible and less technical in the way of targeting the tumour microenvironment.

### 5.3. Peptide Nanoparticle DSSs

Linear and cyclic peptides that are either synthesised or derived from existing fragments of naturally occurring proteins are also important contributors to the nanoparticle DSSs that are currently available. Peptides are often used as the targets for cell surface receptors since most proteins that bind to such receptors do so via a specific fragment in their peptide sequence. These coupled with their ease of synthesis and low immunogenicity makes peptides a useful tool as potential DSSs. Several peptides have been used alone or indeed as part of a surface modification to other nanoparticles for improved drug delivery. Kim et al. [132] used an encapsulated peptidomimetic of epidermal growth factor receptor (EGFR) ligand in a cationic liposome to aid in the inhibition of EGFR signaling in a lung cancer cell line. Somatostatins, a group of peptide hormones that are ligands highly expressed in different cancer cells, have also been conjugated to different anticancer drugs such as doxorubicin, methotrexate, and paclitaxel to enable selective targeting of cancer cells [133]. RIPL peptide is a short peptide that was developed to bind hepsin, a serine protease, which is highly expressed on the surface of hepatoma and prostate cancer cells. Kang et al. [134] developed a RIPL-conjugated liposome to selectively target a panel of cancer cells that overexpresses hepsin, yielding an increased selectivity and cellular uptake compared to conventional liposomes without the targeting peptide.

### 5.4. Inorganic Nanoparticle-Based DSSs

Inorganic nanoparticles have been widely studied for their potential use in drug delivery systems due to their unique properties, such as small size, biocompatibility, and stability. Inorganic nanoparticles, such as dendrimers, and inorganic nanocarriers such as silica, magnetic, and gold nanocarriers, can be used to encapsulate and deliver drugs to specific target sites in the body.

Dendrimers are branched, nanoscale polymers that have attracted significant attention as DSSs due to their unique properties, such as small size, high surface-to-volume ratio, and tunable surface functionality [135]. One of the main advantages of dendrimers as DSSs is their ability to encapsulate a large amount of drug in their interior or on their surface. This allows for controlled and sustained release of the drug, improving its therapeutic efficacy and reducing its side effects. Dendrimers can also be functionalized with targeting moieties, such as antibodies or peptides, which can improve their specificity for a specific target site in the body, such as a tumour cell [136]. Another advantage of dendrimers is their biocompatibility, which is due to their nontoxic and biodegradable nature. This makes them a promising platform for the delivery of a wide range of drugs, including small molecules, proteins, and nucleic acids.

Despite their potential benefits, there are also some challenges associated with the use of dendrimers as DSSs. One of the main challenges is the difficulty in controlling the size and shape of dendrimers, which can affect their pharmacokinetics and biodistribution in the body [137]. In addition, dendrimers can interact with the immune system and cause toxicity, which limits their use in certain applications.

Inorganic nanocarriers, such as silica, magnetic, and gold nanocarriers, have also been used for drug delivery. Silica nanocarriers, for example, can be functionalized with targeting moieties or with a coating that protects the drug from degradation, while magnetic nanocarriers can be directed to specific sites in the body using an external magnetic field [73,105]. Gold nanocarriers have unique optical properties that can be used for imaging and photothermal therapy, making them useful for the treatment of cancer [82].

Inorganic nanoparticles offer a promising approach to drug delivery and have the potential to improve the efficacy and specificity of therapeutic agents. However, more research is needed to fully realize the potential of these nanocarriers and to overcome some of the challenges associated with their use, such as toxicity and stability.

### 5.5. Nanoparticle Delivery Systems and Suppression of Drug-Associated Toxicity

In a bid to improve the activities of conventional drugs where possible, nanoparticles have been the subject of different research in recent years due to their potential pharmacologic properties [138]. Nanoparticles have shown promise in a variety of treatment options ranging from treatment of cancer, kidney disease, neurodegenerative diseases, and even medical imaging for diagnostic purposes [139], and in some instances, surpass the performances of conventional treatment methods. It is now known that the use of nanoparticles alone or by incorporating them into nanocomposites offers improved potential for targeted drug delivery and potentially offers more effective disease treatment [140].

Nanoparticles extensively researched for their medical applications include AgNPs, AuNPs, silicon/silicon oxides and iron oxide nanoparticles. Of these, AgNPs have been extensively researched for medical applications and in fact, AgNP is the most commercialised nanoparticle at present as an active ingredient in an everyday consumable product driven by nanotechnology [141], especially in high concentration. These diverse applications of AgNPs stems from its antibacterial activities and indeed several mechanisms of action have been proposed for their cytotoxic effect. In addition to this, recent investigations have now shifted to investigating the anticancer properties of AgNPs with interesting results. AgNPs have been shown to interact with the DNA inducing DNA damage. AgNPs can also induce ROS which further causes DNA single and double-strand breaks in addition to DNA adducts due to the oxidation of certain nucleotides like guanine to 8-oxo-2-deoxyguanosine, which can base pair with deoxyadenosine resulting in mutation [142,143,144] (Figure 4). This, in addition to the permeabilisation of the mitochondria membrane, can lead to the activation of caspase-dependent cell death. However, the overall effect of both anticancer and antibacterial applications of AgNP possesses increased toxicity risk due to increased and repeated human exposure to the free silver ion (Ag^+^) released into the local environment by the nanoparticle. Ag^+^ released from AgNPs has been documented to cause several side effects such as skin irritation and discolouration, hepatotoxicity, kidney damage, DNA damage, and epithelia cell damage [145].

AgNPs have a dose and size-dependent effect on cellular cytotoxicity, which influence the dynamic changes within the cell. AgNPs can induce apoptosis via the caspase-dependent mitochondrial cell death pathway facilitating cellular dynamics that can damage the cell barrier, inactivate ATPase activity to cause inactivation of Ca^2+^ ATPase and Na^+^/K^+^ ATPase. This, in addition to single and double-strand breaks that is caused by AgNP-induced DNA damage, can excessively generate and accumulate ROS causing the permeabilisation of the mitochondrial membrane and release of cytochrome C and pro-apoptotic protein into the cytoplasm followed by activation of the caspase cascade, and finally apoptosis.

Recent studies have successfully demonstrated that the incorporation of AgNPs inside an lipid-based nano-carrier can help mitigate AgNP-associated inflammation and simultaneously improve the cytotoxicity of the nanoparticle [146,147,148]. Yusuf, Brophy, Gorey and Casey [146] and Yusuf and Casey [147] in their study where THP-1 cells and THP-1 differentiated macrophages were exposed to liposomal AgNPs discovered that the liposome improved the nanoparticle uptake but modified the intracellular signaling activities, induced by the nanoparticle, to improve the nanoparticle cytotoxicity inducing significant cell death at a considerably lower concentration than the uncoated AgNPs. This can be a proof of concept for all nanoparticles or conventional drugs with toxic side effects at high concentrations, where these drugs can be incorporated in lipid carriers for improved delivery and lower concentrations.

## 6. Summary and Future Directions

Undoubtedly, nano-technological advancements have improved the way we view the world, owing to the immense contributions in the field of medicine, food and cosmetics industry, and personal care. These improvements have also been responsible for the expanding global markets of nanotechnology, which thus increases its viability and contribution to economies around the globe. Considering the medical applications, several conventional drugs with toxic side effects such as doxorubicin have been successfully administered in doses within nano-carriers, preventing side effects that will otherwise limit the application of these drugs. In the food industry, nanoparticles such as AgNPs have also contributed immensely to the preservation of food, extending shelf lives and profit margins.

With all these benefits, there are also set backs. Nanoparticles are often very reactive due to their large surface area and surface chemistry [149]. Nanoparticles get into the environment in a variety of ways including household and industrial wastes. This, in addition to the wide application of nanoparticles in everyday consumable products, may result in toxic effects owing to repeated exposure to these nanoparticles. While no toxic effects may have been reported at present about commercialised nanoparticles, continuous human exposure to such nanoparticles either directly from consumed products or indirectly from the environment may result in toxic side effects in the future. Unfortunately, there are very few studies that have investigated the biological (in vivo) and environmental impacts of the myriads of nanoparticles that are currently in the markets. As such, there is need for studies and research into the investigation of the ecotoxicological and biological effects of the exposure to these nanoparticles to humans and all possible faunas. Such studies will be required to be strategic and carefully designed to allow for the differences in variables such as environmental conditions, environmentally relevant nanoparticle concentrations, and type of biological systems, to be considered for decision-making from the findings of such studies.

Another challenge of nanoparticles in biomedicine is their potential for immune recognition and clearance. The immune system can recognize and eliminate foreign particles, including nanoparticles, through a variety of mechanisms, such as phagocytosis by macrophages [131]. Therefore, strategies to minimize immune recognition and clearance of nanoparticles, such as surface modification and targeting, are necessary for effective biomedical applications. Furthermore, the stability and reproducibility of nanoparticles is a crucial challenge that must be addressed for clinical translation. The physicochemical properties of nanoparticles, such as size, shape, and surface charge, can affect their behaviour in biological systems and the efficacy of their applications [132]. Thus, reproducible synthesis and characterization of nanoparticles is essential for their successful translation to clinical settings. Lastly, the regulatory approval process for nanoparticle-based technologies is still under development, and it can be a challenge for researchers and companies to navigate. The regulatory agencies require extensive safety and efficacy testing to ensure the safety of nanoparticle-based products [133].

In summary, the challenges of nanoparticles in biomedicine include toxicity, immune recognition and clearance, stability and reproducibility, and regulatory approval. Addressing these challenges will be critical to the successful translation of nanoparticle-based technologies to clinical settings.

## Figures and Tables

**Figure 2 polymers-15-01596-f002:**
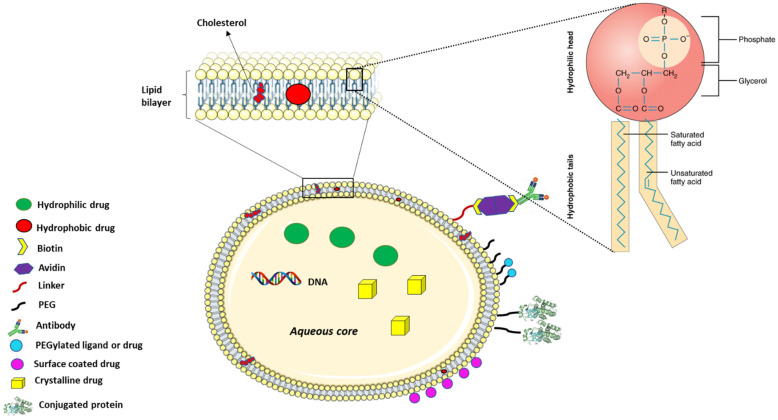
Liposomal modification for drug delivery.

**Figure 4 polymers-15-01596-f004:**
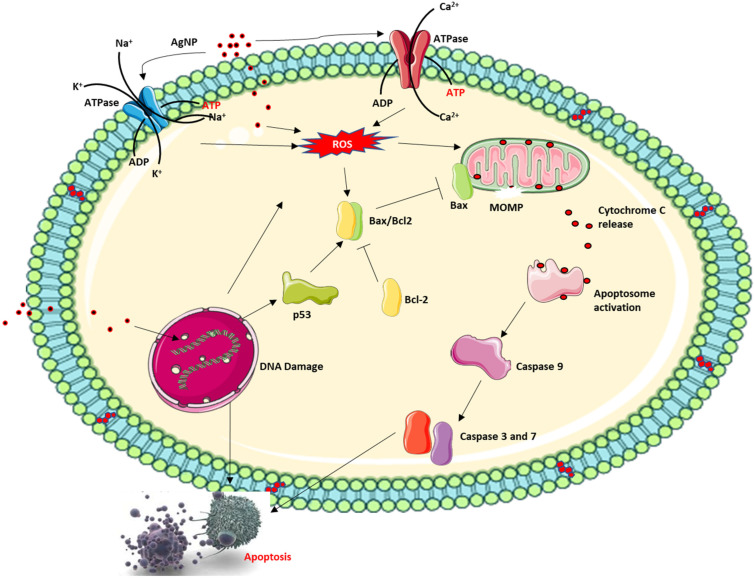
Proposed mechanism of action of AgNPs.

**Table 1 polymers-15-01596-t001:** List of nanodrugs approved by the FDA for clinical application [41,42,43].

	Trade Name (Manufacturer)	Component	Delivery Method	Indication(s)	Approval Year	References
**Liposomal nanoparticle**	AmBIsome (Gilead Sciences, Foster City, CA, USA)	Amphotericin B in liposome	Intravenous infusion	Fungal/protozoal infections	1991	[44]
	Onivyde (Ipsen Biopharmaceuticals, Paris France)	Irinotecan in liposome	Intravenous injection	Metastatic adenocarcinoma of the pancreas	2015	[45]
	Vyxeos (Jazz Pharmaceuticals, Dublin Ireland)	Daunorubicin and cytarabine in liposome	intramuscular, intrathecal, or subcutaneous injection	AML, AML due to previous cancer therapy, AML with myelodysplasia related changes	2017	[46]
	Doxil (Janssen, Beerse Belgium)	Doxorubicin-HCl in liposome	Intravenous infusion	Multiple myeloma Kaposi’s sarcoma and Ovarian cancer	1995	[47]
	Marqibo (Spectrum Pharmaceuticals, Boston, MA, USA)	Vincristine in liposome	Intravenous injection	ALL	2012	[48]
	DepoDur (Pacira Pharmaceuticals, San Diego, CA, USA)	Orphine sulphate in liposome	Epidural administration	Postoperative pain	2011	[49]
	Visudyne (Bausch and Lomb, Vaughan, ON, Canada)	Verteporfin in liposome	Intravenous injection	subfoveal choroidal neovascularization due to AMD, myopia	2000	[50]
**Polymer NPs**	Adagen (Leadiant Biosciences, Gaithersburg, MD, USA)	Pegademase bovine	intramuscular injection	SCID	1990	[51]
	Plegridy (Biogen, Cambridge, MA, USA)	Pegylated IFN-β-1a	Subcutaneous injection	Multiple sclerosis	2014	[41]
	Somavert (Pfizer, New York, NY, USA)	Pegvisomant	Subcutaneous injection	Acromegaly	2007	[52]
	Eligard (Tolmar, Chicago, IL, USA)	Leuprolide acetate and polymer	Subcutaneous injection	Prostate cancer	2003	[53]
	Cimzia (UCB, Brussels, Belgium)	Certolizumab pegol	Pills or intravenous injection	Rheumatoid arthritis, Crohn’s disease, psoriatic arthritis, ankylosing spondylitis	2009	[54]
	Macugen (Bausch and Lomb, Vaughan, ON, Canada)	Pegaptinib	Intravitreal injections	Neovascular AMD	2004	[55]
	Neulasta (Amgen, C Thousand Oaks, CA, USA)	Pegfilgrastim	On-body injection	Chemotherapy-induced neutropenia	2002	[56]
	Pegasys (Genentech, Souht San Francisco, CA, USA)	Pegylated IFN alpha-2a	Subcutaneous injection	Hepatitis B, hepatitis C	2002	[51]
	PegIntron (Merck, Kenilworth, NY, USA)	Pegylated IFN alpha-2b	Subcutaneous injection	Hepatitis C	2001	[57]
	Copaxone (Teva, Tel Aviv Israel)	Glatimer acetate	Subcutaneous injection	Multiple sclerosis	1996	[58]
	Rebinyn (Novo Nordisk, Bagsværd, Denmark)	Coagulation factor IX (recombinant), glycopegylated	Intravenous injection	Haemophilia B	2008	[41]
	Zilretta (Flexion Therapeutics, Woburn, MA, USA)	Triamcinolone acetonide	Intra-articular injection	Osteoarthritis knee pain	2017	[41]
**Micellar NPs**	Abraxane (Celgene, Summit, NJ, USA)	Micellar containing paclitaxel	Intravenous	breast cancer, non-small cell lung cancer, pancreatic cancer, and ovarian cancer	2005	[41]
**Nanocrystal nanoparticles**	Rapamune (Wyeth Pharmaceuticals, Madison, WI, USA)	Sirolimus	Oral administration	Immunosuppressant	1999	[41]
	Emend (Merck, Kenilworth, NJ, USA)	Aprepitant	Oral administration	Antiemetic	2003	[59]
	Focalin (Novartis, Cambridge, MA, USA)	Dexamethylphenidate HCl	Intravenous injection	Psychostimulant	2002	[60]
	Invega Sustenna (Janssen, Beerse Belgium)	Paliperidone palmitate	Intramuscular injection	Schizophrenia, schizoaffective disorder	2009	[61]
	Zanaflex (Acorda, Ardsley, NY, USA)	Tizanidine HCl	Oral administration	Muscle relaxant	1997	[62]
	Ritalin (Novartis, Cambridge, MA, USA)	Methylphenidate HCl	Oral administration	Attention deficit hyperactivity disorder and narcolepsy	1955	[63]
	Avinza (Pfizer, New York City, NY, USA)	Morphine sulfate	Oral administration	Severe pain	2002	[64]
	Tricor (AbbVie, Chicago, IL, USA)	Fenofibrate	Oral administration	Hyper-lipidemia	1998	[41]
**Inorganic nanoparticles**	Dexferrum (American Regent, Shirley, NY, USA)	Iron dextran	Intravenous injection	Iron deficiency	2009	[65]
	Venofer (Vifor Pharma, Zurich Switzerland)	Iron sucrose	Intravenous injection	Iron deficiency in CKD, IBD	2000	[66]
	Ferrlecit (Sanofi-Aventis, Paris France)	Sodium ferric gluconate complex in sucrose	Intravenous infusion	Iron deficiency in CKD	1999	[66]
	Onpattro (Alnylam Pharmaceuticals, Cambridge, MA, USA)	Antisense mRNA to the transthyretin (TTR) gene	subcutaneous injection	caused by hereditary transthyretin-mediated amyloidosis (hATTR)	2018	[67]
	Feraheme (AMAG Pharmaceuticals, Cambridge, MA, USA)	Ferumoxytol	Intravenous infusion	Iron deficiency in CKD	2009	[68]

*AMD—Age related macular degeneration, AML—Acute myeloid leukamia ALL—Acute lymphoblastic leukaemia, CKD—Chronic kidney disease, IBD—Inflammatory bowel disease, SCID—Severe combined immunodeficiency disease*.

**Table 2 polymers-15-01596-t002:** Nanoparticle types, characteristics, benefits, and application.

Nanoparticle type	Characteristics	Benefits	Applications
Liposomes	Spherical structures made of a lipid bilayer that encapsulates drugs and protects them from degradation. They are biocompatible and can be easily functionalized with targeting moieties for a specific delivery.	Can encapsulate hydrophilic and hydrophobic drugs, biodegradable and biocompatible; can target specific tissues	Drug delivery, gene therapy, vaccine delivery
Dendrimers	Highly branched, monodisperse nanoscale polymers that can be used as drug carriers. They are highly customizable, with a range of sizes, surface functionalities, and drug-loading capacities.	Can be designed to have specific sizes and shapes, high drug loading capacity; can target specific tissues	Drug delivery, gene therapy, imaging
Polymeric nanoparticles	Made from biodegradable polymers that can encapsulate drugs and protect them from degradation. They are often used for the sustained release of drugs over a period of time.	Can encapsulate hydrophilic and hydrophobic drugs; can be functionalized with targeting ligands or imaging agents, biocompatible	Drug delivery, gene therapy, imaging
Metal nanoparticles	Including gold nanoparticles (AuNP) and AgNP, have unique optical, electronic, and thermal properties that make them attractive for use in drug delivery. They can be functionalized with targeting moieties for a specific delivery.	Unique optical and magnetic properties; can be functionalized with targeting ligands or imaging agents; biocompatible	Imaging, cancer therapy, biosensors
Solid lipid nanoparticles:	Made of solid lipids and are used to encapsulate hydrophobic drugs. They offer a number of advantages over other types of nanoparticles, including stability, biocompatibility, and improved bioavailability.	Can encapsulate hydrophilic and hydrophobic drugs, biocompatible; can be functionalized with targeting ligands or imaging agents	Drug delivery, cosmetic, and personal care products

## Data Availability

Not applicable.

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
