# Peer review of "Nanoparticles as Drug Delivery Systems: A Review of the Implication of Nanoparticles’ Physicochemical Properties on Responses in Biological Systems"

_polymers, 2023, doi:10.3390/polym15071596_

Round 1

Reviewer 1 Report

This is a great article. Contains in a concise form very valuable and up-to-date information. I have only one remark. 1) Figure 3 must have captions, i.e. the name of the processes, the molecules that are delivered to the cell, etc.

Reviewer 2 Report

The authors have done a literature survey on improve biological responses of NPs and Nano-carriers. The manuscript is organized well and could be of an interest to a broad range of readers of this journal. However, there are a few issued that need to be addressed before suggestion for publication.

-          It is recommended to add an introduction explaining the state of the art in the field of nanotechnology.

-          The title of this review paper suggest a specific scope of biological response for nanoparticles and nano-carries, whereas the manuscript is covering a broader application such as food industry etc. Could the author either revise the title to better reflect the scope of this manuscript or the sections provided.

-          There is no clear classification on type/design of the Nanoparticle/carrier used in each of the sectors mentioned in this review paper.

-          it is recommended to provide more than one citation of statement made in the manuscript; no example of other types of nanoparticles i.e. inorganic ones have been provided (doi.org/10.1021/acsami.6b14836)

-          could the author add references to the table 1?

-          The use of nanotechnology in cancer diagnostic need to be elaborated more, methods such as Raman spectroscopy in early cancer diagnosis is missing (doi.org/10.1002/smtd.202200798)

-          please provide more examples in the section 3.1 size and area (doi.org/10.1002/prca.200900077)

Reviewer 3 Report

In the manuscript “Nanotechnology and nanoparticle design: A review of nanoparticles and nano-carriers for improved biological responses”, authors describe the general development of nanotechnology, the characteristics of nanoparticles as well as their applications in medicine and industry. Overall, the scheme of the manuscript is interesting, however, the review in its current form lacks the contents and needed to be suitable for publication in Polymers because of the following points.

1.     In the section 2.1 and 2.2, the authors claimed the application of nanotechnology in Food industry and Cosmetic industry. However, few examples were provided to show the usage of nanotechnology in these industries. It would be recommended to use table/figure to illustrate the properties, benefits, and applications of nanoparticles.

2.     Some sentences need to be revised to make them clearer to the readers, such as page 3, line 127-132.

3.     In section 2.3, the authors focus on elaborating the application of nanotechnology in medicine. However, the overall structure is chaotic and it doesn’t provide much information for the biomedical application of nanoparticles. It would be advisable to separately discuss in subsections, instead of mixing everything in one paragraph. For instance, in page 3, line 139-142, the authors briefly discussed the therapeutic application of nanoparticles. What is confused to the readers is, whether the authors want to elaborate the therapeutic molecules inside the nanoparticles or nanoparticles themselves?

4.     In section 2.3, the authors mentioned the advantages of nanoparticles-based DDS in penetrating cell membrane and targeting to disease area. Please also include this information in Table 1 to clearly show the properties and mechanisms.

5.     Please add the year of approval in Table 1.

6.     siRNA/mRNA based nanoformulation approved by FDA is missing in Table 1.

7.     Some sentences need to be revised to make them scientifically sound, such as page 6, line 188-190; page 6, line 216-218; page 8, line 300-302.

8.     In line 164-190, the authors discuss on the imaging applications of iron/silica-based nanoparticles. Please give more detailed information to show the properties. For examples, the current application of these nanoparticles in clinic, the mechanisms for biomedical imaging, the major concerns/parameters need to consider when design these types of nanoplatform.

9.     Figure 1 isn’t representative enough to show the physicochemical properties of nanoparticles, please replace with an appropriate one.

10.  In section 3.1, little information was given to discuss the mechanism of intracellular localization of NPs.

11.  Please use the subscript of some letters, such as ZnO2.

12.  In Page 9, line 320-321, what do the authors mean with “biomedical importance”?

13.  In section 3.3, it doesn’t provide much information on the mechanisms of different shapes-affected endocytosis of NPs.

14.  For section 5, it would be recommended to include a subsection to illustrate the inorganic NPs based DDS.

15.  In figure 3, please give some indications to clearly show the cellular uptake mechanism of liposome-based DDS.

16.  It would be advisable to make a table to more clearly reflect the characteristics of different types of NPs as drug delivery systems.

17.  What is the main idea of section 6? If the authors want to discuss more on the reduced cytotoxicity by applying NPs as DDS, it can be included in section 5, instead of an independent section.

18.  For the figures used in this review, are they drawn by the authors or quoted from the literatures? If so, please add the references in figure legends.

19.  The title of this review needs to revise to be consistent with the contents.

20.  In general, this review in its current form lacks the contents, more detailed information should be provided to publish in this journal.
